# The Endosperm-Specific Gene *OsEnS-42* Regulates Seed Vigor and Grain Quality

**DOI:** 10.3390/plants14162492

**Published:** 2025-08-11

**Authors:** Minhua Zheng, Xiaodan Hu, Luo Chen, Jiale Xing, Shuai Nie, Lukai Ma, Wei Sun, Dilin Liu, Xiumei Li, Weerachai Matthayatthaworn, Wu Yang, Wei Liu

**Affiliations:** 1College of Agriculture and Biology, Zhongkai University of Agriculture and Engineering, Guangzhou 510225, China; zheng1802040210@163.com (M.Z.); 13229416940@163.com (L.C.); malukai@zhku.edu.cn (L.M.); starking521@126.com (W.S.); 2Rice Research Institute, Guangdong Academy of Agricultural Sciences, Guangdong Key Laboratory of New Technology in Rice Breeding, Guangdong Rice Engineering Laboratory, Key Laboratory of Genetics and Breeding of High Quality Rice in Southern China (Co-Construction by Ministry and Province), Guangzhou 510640, China; xiaodanhu1827@163.com (X.H.); 17839360557@163.com (J.X.); nieshuai@gdaas.cn (S.N.); dilin_liu@163.com (D.L.); 3Guangdong Provincial Key Laboratory for Crop Germplasm Resources Preservation and Utilization, Agro-Biological Gene Research Center, Guangdong Academy of Agricultural Sciences, Guangzhou 510640, China; lixiumei@agrogene.ac.cn; 4Department of Agronomy, Faculty of Agriculture, Kasetsart University, Nakhon Pathom 73140, Thailand; fagrwhm@ku.ac.th

**Keywords:** rice, seed vigor, grain quality

## Abstract

Seed vigor critically determines sowing performance, while grain quality fundamentally influences commercial value. Elucidating the genetic mechanisms governing these traits is critical for enhancing both seed vigor and grain quality in rice cultivation. Here, we demonstrate that the endosperm-specific gene *OsEnS-42* is highly expressed in germinating seeds and developing seeds at the early filling stage. OsEnS-42 is localized in the nucleus and cytoplasm. The seed vigor of *OsEnS-42* knockout plants decreased, manifested as decreases in germination rate, seedling length, and root length. In addition, *OsEnS-42* knockout plants showed increased chalkiness and amylose content. The transcriptome and physiological indicators showed that *OsEnS-42* regulates seed vigor through soluble sugars and redox metabolism, and regulates grain quality via soluble sugars and seed development-related enzymes. Haplotype analysis of *OsEnS-42* across global rice germplasm revealed four distinct haplotypes (Hap 1–4) with subspecies-specific distributions. Crucially, accessions with Hap 4 exhibit a lower percentage of grain with chalkiness than accessions with Hap 1 (predominantly *indica*), enabling marker-assisted introgression to reduce chalkiness without subspecies barriers. Meanwhile, accessions with Hap 2 show lower amylose content, providing targets for specialty rice breeding. Our findings elucidate the pathways through which *OsEnS-42* regulates seed vigor and grain quality, and provide new molecular breeding targets for improving seed vigor and grain quality in rice.

## 1. Introduction

Rice (*Oryza sativa* L.) is one of the most important cereal crops in the world. With the increasing frequency of extreme environments, improving rice yield and quality has become a top priority. Seed vigor and grain quality are two key traits that affect rice yield and commercial value, and they are both complex traits regulated by genetics and environment [1,2].

Seed vigor, representing the physiological competence for rapid germination and stress resilience, is an important indicator of seed sowing quality, manifested by traits such as rapid germination, stress resistance during germination, and quick establishment of seedlings. Seed vigor is influenced by seed development, maturation, aging, germination, and environmental factors [1]. Recent advances in QTL mapping, genome-wide association studies (GWAS), and multi-omics approaches have identified numerous genes regulating seed vigor, primarily involving hormone signaling, amino acid metabolism, reactive oxygen species (ROS) homeostasis, and energy metabolism. Abscisic acid (ABA), gibberellic acid (GA), IAA, BR, JA, and ET have been reported to be involved in seed vigor, and a number of genes regulate seed vigor through various hormone pathways. For example, *OsIAGLU* [3] and *OsHIPL1* [4] regulate seed vigor via ABA metabolism and signaling. *OsSAP8* [5] promotes seed germination by enhancing GA biosynthesis. The hydrolysis and metabolism of stored substances in seeds are the basis for seed germination. *OsIPMS* provides more energy for seed germination and seedling growth by increasing the biosynthesis of free amino acids during rice seed germination [6]. *OsPK5* regulates glycolysis during seed germination [7]. *OsPFP1* [8] and the PcG-OsFIE1 complex [9] affect seed vigor by modulating storage substance accumulation. As redox signaling mediators, ROS regulate seed germination through oxidative modification [10,11]. Their spatiotemporal dynamics directly modulate peroxidase activity and cell wall loosening, explaining why low levels promote germination while excess causes damage. This concentration-dependent duality establishes ROS homeostasis as a pivotal vigor determinant [12]. *OsRACK1A* [13], *OsPAO5* [14], *OsCDP3.10* [15], and *OsPER1A* [16] regulate seed vigor by modulating ROS homeostasis within seeds.

Grain quality refers to the various basic characteristics that a commodity possesses throughout the entire process from rice production to processing into a consumable product [17]. Grain quality includes milling quality, appearance quality, cooking and eating quality, and nutritional quality. The rice endosperm accumulates storage substances mainly composed of starch and protein during the grain filling stage, ultimately shaping the grain quality and providing energy for seed germination [18]. Currently, some studies indicate that endosperm-specific genes (*OsEnS*) play significant roles in grain quality. The knockout of *OsNF-YB1* (*OsEnS-41*) delays grain filling, reduces grain size, and increases chalkiness [19,20]. *OsbZIP76* interacts with *OsNF-YB1* (*OsEnS-41*) and *OsNF-YB9* (*OsEnS-83*) to modulate yield and quality [21]. The deficiency of *OsNF-YC12* (*OsEnS-133*) reduces protein and starch contents, disrupts starch granule structure, and increases chalkiness [22]. In addition, two endosperm-specific genes have been reported to play roles in seed vigor. *OsEnS-57*, encoding a cupin domain protein characterized by a conserved β-barrel fold, can improve rice seed germination under salt stress [23]. The knockout of *OsEnS-100* reduces plant height, panicle length, and grain weight during the maturation period of rice, while also leading to decreased seed vigor [24].

Grain quality is fundamentally determined by the accumulation of storage compounds, which concurrently govern seed germination capacity. Consequently, seed vigor exhibits a close association with grain quality, suggesting that overlapping regulatory pathways coordinate these traits. Simultaneously, some findings indicate that distinct regulatory pathways are associated with seed vigor and grain quality [1,17,18]. Although a certain number of genes related to the regulation of seed vigor and grain quality have been reported, many of their regulatory pathways remain unknown; especially, those investigating the synergistic regulation of seed vigor and grain quality have been poorly reported.

In a previous transcriptome study on the *OsEnS* gene family [25], *OsEnS-42* (*endosperm-specific gene 42)*, as a glycoprotein hydrolase gene, is specifically highly expressed in developing seeds at the early filling stage. In this study, we reported for the first time the biological functions of *OsEnS-42* in regulating seed vigor and grain quality, and analyzed the regulatory pathways. Furthermore, we analyzed the allelic variations of *OsEnS-42* and their impacts on grain quality. This study enriches the regulatory mechanisms of *OsEnS* family genes and provides new insights into improving rice seed vigor and grain quality.

## 2. Results

### 2.1. Expression Analysis and Subcellular Localization of OsEnS-42

We investigated the expression patterns of *OsEnS-42* in different tissues during various stages of rice growth (Figure 1A). The results revealed dynamic variations in *OsEnS-42* expression levels, showing a tissue-specific and growth stage-dependent pattern. The *OsEnS-42* gene was mainly expressed at various stages of seed germination and developing seed at the early filling stage (one week after flowering), indicating that *OsEnS-42* may regulate seed germination and seed development.

To elucidate the subcellular localization of OsEnS-42, we constructed the OsEnS-42-GFP fusion protein, which was transiently expressed in rice protoplasts under the control of the CaMV 35S promoter. The mCherry nucleus, which was used as a nucleus marker, was co-transfected with OsEnS-42-GFP. The fluorescence signals of the OsEnS-42-GFP fusion protein overlapped with the nucleus marker and were also widely localized in the cytoplasm (Figure 1B).

### 2.2. OsEnS-42 Regulates Seed Vigor and Grain Quality

To investigate the function of *OsEnS-42* in regulating seed vigor and grain quality, CRISPR/Cas9 gene editing technology was used to generate gene knockout (KO) plants of *OsEnS-42*. Two sequence-specific single guide RNAs (sgRNAs) were designed in Exon 3 and introduced into the wild type ‘ZH11’. Two Cas9-positive lines (*ens-42-1* and *ens-42-2*) were obtained, each with distinct mutations, with *ens-42-1* having a deletion of one bp in the first target and *ens-42-2* having one bp insertion in the first and second target, respectively (Appendix A). Both mutations caused premature termination during OsEnS-42 protein translation (Appendix A).

The seed vigor of *OsEnS-42* knockout lines decreased, manifested by markedly decreased germination rate, shoot length, and root length (Figure 2A–E). Meanwhile, the knockout lines exhibited an increased percentage of grain with chalkiness (PGWC) and degree of endosperm chalkiness (DEC) (Figure 2F–H), along with elevated amylose content (AC), while there was no difference in protein and starch content (Figure 2I–K).

### 2.3. OsEnS-42 Regulates Soluble Sugar Metabolism and Redox Homeostasis During Seed Germination

To investigate the molecular mechanism of *OsEnS-42* in regulating seed vigor, we performed transcriptome sequencing to analyze differentially expressed genes (DEGs) in the ZH11 and KO plants. Comparative analysis revealed 34 down-regulated and 343 up-regulated genes in KO plants (Figure 3A,B). The molecular function of the DEGs was associated with hydrolase activity, hydrolyzing O-glycosyl compounds, copper ion binding, nickel cation binding, modified amino acid binding, and chitin binding. The enrichment analysis of cell component indicated a correlation with the cell wall, external encapsulating structure, extracellular region, vesicle membrane, and retrotransposon nucleocapsid. For the biological process, the DEGs were mainly associated with seed germination, cold acclimation, response to amino acid, 1-aminocyclopropane-1-carboxylic acid, and aminoglycan catabolic process (Figure 3C).

KEGG pathway analysis showed that the DEGs in KO plants are related to pathways such as plant hormone signal transduction, phenylpropanoid biosynthesis, amino sugar and nucleotide sugar metabolism, diterpenoid biosynthesis, MAPK signaling pathway, oxidative phosphorylation, monoterpenoid biosynthesis, cysteine and methionine metabolism, pentose and glucuronate interconversions, and glycerolipid metabolism (Figure 3D). The pathways involved in these DEGs suggest that *OsEnS-42* may regulate seed germination through glucose metabolism, hormones, and redox homeostasis. Notably, among DEGs across these pathways, we identified three established seed vigor-associated genes: *LOC_Os01g18860* and *LOC_Os05g49730* (implicated in redox homeostasis), and *LOC_Os08g36910* (involved in glucose metabolism) (Appendix A and Appendix A).

To further demonstrate the pathways through which *OsEnS-42* regulates seed germination, physiological indices, including 13 soluble sugars, 15 hormone levels, and 7 redox enzymes activity, were measured. The results showed that the contents of glucose (Glc), fructose (Fru), and maltose (Mal), as well as the activity of peroxidase (POD) and ascorbate peroxidase (APX), decreased, while the activity of lipoxygenase (LOX), an enzyme known to inhibit seed germination [26], increased in *OsEnS-42* KO plants (Figure 3E,F). The contents of other soluble sugars (sucrose (Suc), trehalose (Tre)), the activity of redox enzymes (superoxide dismutase (SOD), catalase (CAT), and glutathione reductase (GR)), as well as the malondialdehyde (MDA) content and hormone levels, showed no differences (Appendix A). In addition, eight soluble sugars (fucose (Fuc), rhamnose (Rha), arabinose (Ara), galactose (Gal), lactose (Lac), stachyose (Sta), mannose (Man), and raffinose (Raf)) were not detected. These findings indicate that *OsEnS-42* influences soluble sugar and redox metabolism during the seed germination stage.

### 2.4. OsEnS-42 Influences Soluble Sugar Content and Amylase Activity During Grain Filling

To elucidate the molecular mechanism underlying *OsEnS-42*-mediated regulation of grain quality, transcriptome sequencing was performed to identify DEGs between ZH11 and KO plants. In KO plants, 131 down-regulated DEGs and 60 up-regulated DEGs were observed (Figure 4A,B). In molecular function enrichment analysis, the DEGs were enriched in ATPase activity, enzyme activator activity, transcription regulator activity, chaperone binding, and DNA-binding transcription factor activity. Cell component enrichment analysis indicated that the DEGs were enriched in cell cortex part, cell cortex, exocyst, nucleus, and nuclear part. Biological process analysis showed that the DEGs were associated with glycoprotein biosynthetic process, glycoprotein metabolic process, protein glycosylation, regulation of growth, and growth (Figure 4C).

KEGG pathway analysis showed that the DEGs in KO plants were related to pathways such as ascorbate and aldarate metabolism, biosynthesis of secondary metabolites, lysine degradation, metabolic pathways, galactose metabolism, spliceosome, starch and sucrose metabolism, endocytosis, glutathione metabolism, and protein processing in endoplasmic reticulum (Figure 4D). Integrating the pathways involved in DEGs, *OsEnS-42* may regulate grain quality through starch and sugar metabolism. Intriguingly, we detected two DEGs (*LOC_Os03g31300* and *LOC_Os04g33740*) in the pathways known to regulate grain quality (e.g., chalkiness) by affecting starch structure during grain filling (Appendix A and Appendix A).

The physiological indices, including the contents of soluble sugars and the activity of amylases and redox enzymes, were measured. The results demonstrated that the contents of fructose and glucose decreased, while the contents of sucrose increased (Figure 4E). The activity of total amylase, α-amylase, β-amylase, and key redox enzymes, including glutathione reductase (GR), glutathione S-transferase (GST), Na^+^/K^+^-ATPase, and Ca^2+^/Mg^2+^-ATPase, were reduced in the KO plants (Figure 4F–H). These findings indicate that *OsEnS-42* regulates the contents of soluble sugars and enzyme activity related to seed development during the grain filling stage.

### 2.5. Haplotype Analysis of OsEnS-42

To investigate the genetic diversity of *OsEnS-42*, we analyzed the variations in its promoter region (1000 bp upstream of ATG), coding sequence (CDS), and 3′-untranslated region (3′-UTR) across the core germplasm [27]. A total of 10 variants (3 in the promoter region, 6 in the CDS, and 1 in the 3′-UTR) were identified, defining 4 evolutionarily distinct haplotypes (Hap1–4) (Figure 5A). The haplotype network revealed substantial genetic distances among haplotypes, with Hap 1 and Hap 4 positioned intermediately between Hap 2 and Hap 3, indicating clear differentiation among rice subgroups (Figure 5B). Subspecies-specific differentiation was pronounced. In the core germplasm, Hap 1 (92.5% *indica*) and Hap 4 (97.9% *indica*) predominated in *indica*, while Hap 2 (90.7% *japonica*) and Hap 3 (94.5% *aus*) showed near-fixation in their respective subgroups (Figure 5C). This pattern persisted in the 3K rice genomes: Hap 1 (96.1% *indica*) and Hap 4 (96.8% *indica*) remained indica-specific, whereas Hap 2 (45% *indica*, 50% *japonica*) and Hap 3 (60.1% *aus*, 38.3% *indica*) exhibited admixed distributions suggestive of differential selection pressures (Figure 5D). Based on the phenotype data of chalkiness [28] and AC [29] reported by the core germplasm, we conducted association analysis between the four haplotypes and phenotypes. For chalkiness, accessions with Hap 1 showed higher PGWC than that of accessions with Hap 4 across the two environments. For AC, accessions with Hap 2 showed lower AC than that of accessions with the other three haplotypes across the two environments (Figure 5E,F).

## 3. Discussion

Seed vigor and grain quality are key factors affecting rice yield and commodity value, governed by the accumulation and metabolism of grain storage substances. The grain quality largely depends on the accumulation of storage substances, which simultaneously determines seed germination capacity [1,30], establishing a physiological link between these traits. The members of the *OsEnS* family can be divided into 12 categories, including transcription factors, stress/defense, seed storage proteins, carbohydrate and energy metabolism, seed maturation, protein metabolism, lipid metabolism, transport, cell wall related, hormone-related, signal transduction, and an unclassified group, widely involved in various plant growth and development processes [25]. Our findings suggest that *OsEnS-42* performs distinct roles from previously characterized members. Unlike transcriptional regulators *OsNF-YB1* (*OsEnS-41*), *OsNF-YC12* (*OsEnS-133*), and *OsNF-YB9* (*OsEnS-83*), which modulate starch synthesis [19,20,22], or *OsEnS-57*/*OsEnS-100* affecting stress responses and vigor [23,24], *OsEnS-42* may function as a glycoside hydrolase regulating sugar metabolism. Our data indicate that *OsEnS-42* regulates seed germination by affecting sugar metabolism and redox processes (Figure 3). Specifically, the increase in LOX activity in knockout lines may drive lipid peroxidation and ROS overaccumulation based on established mechanisms [26], potentially causing oxidative damage to cellular membranes and organelles. This could impair imbibition efficiency and energy metabolism during germination—a key contributor to reduced vigor. Moreover, *OsEnS-42* appears to regulate grain quality by adjusting soluble sugar content and enzyme activity related to seed development (Figure 3 and Figure 4). Although storage substance accumulation is the foundation of grain development and maturation, the energy required for grain development, such as energy provided by respiration, also requires the consumption of storage substances. A better balance of energy and metabolism is the key to improving grain quality [31]. The metabolism and pathways of some major energy metabolites, such as sucrose, can affect grain quality [32]. The temporal expression peak during early filling (Figure 1) coincides with this energy-critical phase, where KO lines show the following: (1) decreased monosaccharides (Glc/Fru) essential for cellular energy, (2) increased sucrose accumulation, and (3) reduced ATPase activity (Figure 4). This is consistent with *OsEnS-42*’s role in providing metabolic energy for storage compound deposition rather than directly controlling starch/protein levels, as evidenced by unchanged protein and total starch content in KO lines despite elevated amylose (Figure 2).

The air spaces created by incomplete grain filling are the main cause of chalkiness. Concurrently, additional factors contribute to chalkiness formation, including alterations in amylose content, starch structural modifications, and variations in protein content [29,33]. The knockout lines of OsEnS-42 exhibited increased amylose content, while there was no difference in protein and starch content (Figure 2I–K). The impaired sucrose-to-monosaccharide conversion in KO plants reduces hexose availability during early filling, creating an ADP-glucose limited state that favors amylose synthesis over amylopectin. Furthermore, the altered sugar metabolism may modify starch granule initiation, where spatial constraints in developing granules could independently affect amylose incorporation without changing total starch accumulation. The changes in starch structure caused by sugar metabolism may ultimately lead to the formation of chalkiness.

Four haplotypes of *OsEnS-42* were identified in this study. Hap 1 and Hap 4 were predominantly found in the *indica* subgroup. Interestingly, across both environments, accessions with Hap 4 exhibited a lower percentage of grain with chalkiness than accessions with Hap 1 (Figure 5E,F). This provides informative allelic variations for breeders to introduce Hap 4 into accessions with Hap 1 to reduce chalkiness without considering cross incompatibility barriers between subspecies. Although there is a high correlation between AC and chalkiness, there is no stable difference in chalkiness between accessions with Hap 2 (lower AC) and accessions with Hap 1, Hap 3, or Hap 4, indicating the existence of different regulatory pathways between AC and chalkiness. It is worth noting that the AC of accessions with Hap 2 was lower than that of accessions with the other three haplotypes (Figure 5E,F), which is likely due to the large proportion of *japonica* accession in Hap 2, as the AC of *japonica* rice is often significantly lower than that of *indica* rice [29]. It is worth further analysis and research to determine whether the differences in AC content are caused by inter-subspecies or possibly due to allelic variations of *OsEnS-42*. In general, the pathways through which *OsEnS-42* regulates seed vigor and grain quality, as well as the allelic variations of *OsEnS-42*, provide valuable information for further understanding the genetic basis of seed vigor and grain quality.

## 4. Materials and Methods

### 4.1. RNA Extraction and qRT-PCR

The target gene (MSU_Locus: *LOC_Os02g54960*) in this study was named *OsEnS-42* by the reported research [25]. To analyze the expression pattern of *OsEnS-42*, the total RNA was extracted from various tissues of ZH11 plants using Trizol reagent (Takara, Dalian, China) following the manufacturer’s protocol. Approximately 500 ng of total RNA was reverse-transcribed into first-strand cDNA using the Evo M-MLV Reverse Transcription Kit (AccuraBio, Shanghai, China). qRT-PCR was performed with SYBR Green Pro Taq HS Premix (AccuraBio, China) on a Roche LightCycler 480 II Real-Time PCR System (Roche, Basel, Switzerland). The *eEF1α* gene was used as an internal control. The primer sequences of *eEF1α* are as follows: forward primer: 5′-TTTCACTCTTGGTGAAGCGAT-3′; reverse primer: 5′-GACTTCCTTCACGATTTCATCGTAA-3′. The primer sequences of *OsEnS-42* are as follows: forward primer: 5′-CAACAACTGGTAGGAATCA-3′; reverse primer: 5′-GACCCAGCACCTATCCCAAC-3′. Three biological replicates were included.

### 4.2. Subcellular Localization of OsEnS-42

The full-length coding sequence of *OsEnS-42* was cloned into the Pegoep35S vector, which harbors a green fluorescent protein (GFP) tag driven by the CaMV 35S promoter. The recombinant plasmid and a control vector were co-transfected into rice protoplasts according to the reported study [34]. Briefly, the dehulled rice seeds of ZH11 were surface-sterilized in 1.5% sodium hypochlorite (40 min), then germinated in darkness on half-strength MS medium for 7 days. Protoplasts were isolated via enzymatic digestion (1.5% cellulase, 0.75% macerozyme). For transient expression, protoplasts were transfected with 1 μg plasmid DNA for 24 h incubation at 28 °C without light. The fluorescence signals were visualized and captured using a confocal laser scanning microscope (Olympus FV1000, Tokyo, Japan).

### 4.3. Construction of OsEnS-42 Knockout Lines

The transgenic vector was constructed using the CRISPR/Cas9 system. The target sequences were as follows: Target 1: 5′-AAGCTCTTCCCCCATAAGGT-3′; Target 2: 5′-CTGGACACCACGATGGTGCA-3′. The knockout vector pEGCas9Pubi-H-*OsEnS-42*-ko was transformed into Agrobacterium tumefaciens strain EHA105 and subsequently used to infect callus tissues of wild type ‘ZH11’. Homozygous transgenic lines were obtained through seed propagation and segregation. Positive lines of gene editing were detected by primers (forward primer: 5′-CCAAAACCCGCATATCATTGA-3′; reverse primer: 5′-AACTGGTCCTACCAATGGACAC-3′) and compared with the ZH11 genome.

### 4.4. Seed Vigor Measurement

Mature seeds of ZH11 and two knockout lines (*ens-42-1* and *ens-42-2*) were air-dried naturally. Plump and healthy seeds were selected and placed in 9 cm diameter dishes with distilled water. The seeds were soaked in water (fully submerged) for 24 h, and subsequently germinated under moist conditions (without submergence). This experiment was conducted in a climate chamber at 28 ± 2 °C under a 12 h light/dark cycle. Germination rates were recorded every 12 h until no further germination was observed. A seed was considered germinated when the radicle penetrated the seed coat by 2 mm [7]. Root and shoot lengths were measured after 96 h under germination conditions. This experiment was conducted in three biological replicates, with at least three parallel experiments per replicate, each containing 50 seeds.

### 4.5. Grain Quality Analysis

Mature seeds of ZH11 and the knockout lines (*ens-42-1* and *ens-42-2*) were air-dried and processed using a rice husker (JLG-III, Sinograin, China) to obtain brown rice. Brown rice was polished into white rice using an experimental polishing machine (NA-JCB, Beijing Jinbiao Guoan Technology Co., Ltd., Beijing, China). For chalkiness assessment, 100–200 whole polished grains were evenly arranged on a scanner and analyzed with a grain appearance quality analyzer (SC-E, Hangzhou Wanshen Testing Equipment Co., Ltd., Hangzhou, China) to determine DEC and PGWC. Three independent replicates were conducted.

For the determination of total starch, polished rice was ground into flour and treated with anhydrous ether under agitation. After centrifugation, the supernatant was discarded. Ethanol was added to homogenize the residue, followed by water bath incubation and centrifugation. The pellet was resuspended in distilled water, gelatinized in boiling water, and cooled. Hydrochloric acid (HCl) was added for hydrolysis, and the mixture was centrifuged. The supernatant was diluted, neutralized with NaOH, and mixed with 3,5-dinitrosalicylic acid (DNS) reagent. After boiling and cooling, absorbance at 540 nm was measured to calculate total starch content. Three experimental replicates were performed. The AC was determined according to the reported methods [29].

For the measurement of total protein content, samples were homogenized and combusted in a high-purity oxygen atmosphere (≥99.99%) at 900–1200 °C using a combustion reactor. Combustion products were transported by carrier gas (CO_2_ or He) to a reduction furnace (800 °C) and converted to nitrogen gas. Nitrogen content was quantified using a DN3000 Dumas analyzer (Beijing Nuode Instrument Co., Ltd., Beijing, China). The total protein content was 6.25 times the nitrogen content in the sample. Three replicates were performed.

### 4.6. Transcriptome Sequencing

The seeds were sampled after 48 h under germination conditions and panicles were sampled on the 7th day after flowering. The total RNA was extracted using Trizol reagent. Sequencing was conducted on a second-generation high-throughput sequencing platform. DEGs were identified using DESeq2 with thresholds of |log_2_ (fold change) | > 1 and false discovery rate (FDR) < 0.05. DEGs were further analyzed via Gene Ontology (GO) enrichment and Kyoto Encyclopedia of Genes and Genomes (KEGG) pathway analyses.

### 4.7. Measurement of Physiological Indicators

Superoxide dismutase (SOD) activity: samples were ground into powder with liquid nitrogen and homogenized in phosphate-buffered saline (PBS) on ice. After centrifugation, the supernatant was mixed with PBS, xanthine oxidase, nitro blue tetrazolium (NBT), distilled water, and xanthine solution. Absorbance at 560 nm was measured after incubation, and SOD activity was calculated. Three replicates were included. Peroxidase (POD) activity: Homogenized samples (as above) were centrifuged, and the supernatant was mixed with PBS. Absorbance at 470 nm was recorded at 30 s and 330 s after mixing. POD activity was calculated using the absorbance difference. Three replicates were conducted. Catalase (CAT) activity: The supernatant from homogenized samples was mixed with H_2_O_2_ and incubated. Ammonium molybdate tetrahydrate was added, and absorbance at 405 nm was measured after 10 min at room temperature. CAT activity was calculated. Three replicates were performed. Lipoxygenase (LOX) activity: Samples were homogenized in sodium phosphate and acetate buffers containing sodium linoleate. After centrifugation, the supernatant was mixed with acetate buffer and sodium linoleate. Absorbance at 234 nm was recorded at 15 s (A_1_) and 75 s (A_2_). LOX activity was calculated. Three replicates were included. Malondialdehyde (MDA) content: Homogenized samples were centrifuged, and the supernatant was mixed with thiobarbituric acid (TBA). After heating and cooling, absorbance at 532 nm and 600 nm were measured. MDA content was calculated. Three replicates were conducted.

Ascorbate peroxidase (APX) activity: The supernatant from homogenized samples was mixed with PBS, ascorbic acid (VC), and H_2_O_2_. Absorbance at 290 nm was recorded at 10 s (A_1_) and 610 s (A_2_). APX activity was calculated. Three replicates were performed. Glutathione reductase (GR) activity: The supernatant was mixed with NADPH and oxidized glutathione (GSSG) in PBS. Absorbance at 340 nm was measured at 10 s (A_1_) and 310 s (A_2_). GR activity was calculated. Three replicates were included.

Hormone content analysis: Samples were extracted with acetonitrile/water, sonicated, and centrifuged. The supernatant was purified using reversed-phase solid-phase extraction (RP-SPE) columns. Eluted fractions were dried under nitrogen, reconstituted, and analyzed using a Vanquish UPLC system coupled to a Q Exactive high-resolution mass spectrometer (Thermo Fisher Scientific, USA) [35,36,37,38]. Three replicates were performed.

Soluble sugar content: Samples were extracted with ethanol, diluted, and analyzed using an ICS 5000+ ion chromatography system (Thermo Fisher Scientific, Waltham, MA, USA) with an electrochemical detector [39,40,41]. Three replicates were conducted.

Total amylase activity: Homogenized samples were incubated with starch solution at 37 °C for 10 min. Iodine solution was added, and absorbance at 660 nm was measured. Total amylase activity was calculated. α-amylase activity: Samples were heated at 70 °C to inactivate β-amylase before starch incubation. Activity was measured as above. β-amylase activity: Calculated as total amylase activity minus α-amylase activity. Three replicates were performed for each assay.

### 4.8. Haplotype Analysis

Insertion/deletion (InDel) and single nucleotide polymorphism (SNP) variations in *OsEnS-42* were analyzed using resequencing data from the core germplasm [27] and the 3K rice genome project (https://mbkbase.org/rice/genotype, accessed on 20 December 2024). The *Nipponbare* genome was used as the reference. Haplotype networks were constructed using the geneHapR [42].

### 4.9. Data Processing

Phenotypic data and variance were analyzed using Microsoft Excel 2003. Significant differences were determined by Student’s *t*-test at *p* < 0.05 and *p* < 0.01 levels.

## Figures and Tables

**Figure 1 plants-14-02492-f001:**
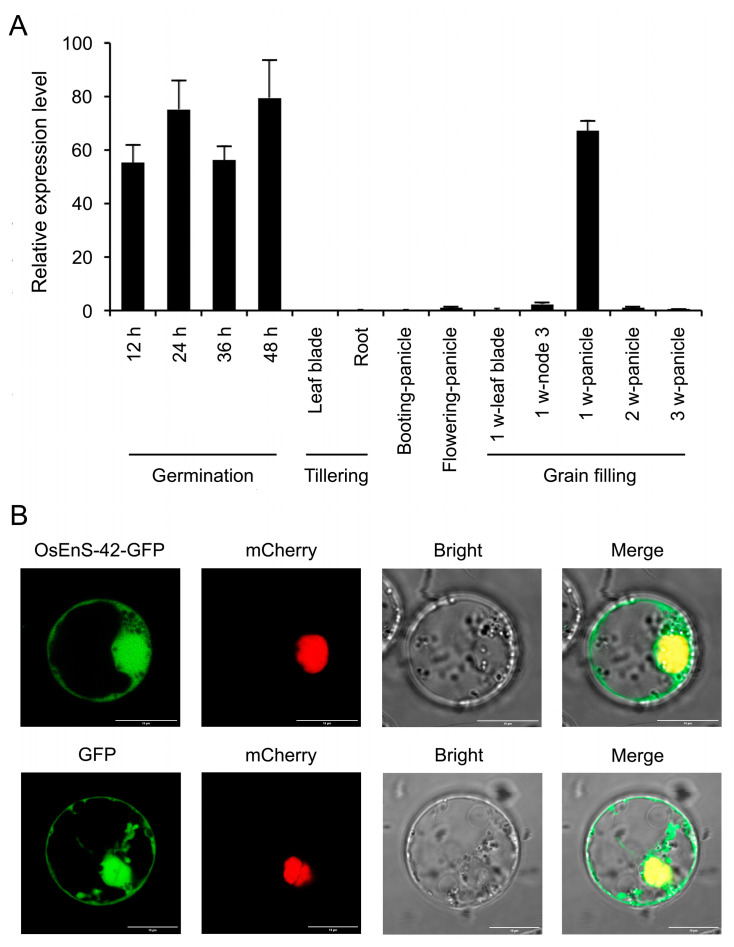
Expression analysis and subcellular localization of OsEnS-42. (**A**) Relative expression level of *OsEnS-42* in various tissues at different growth stages. Rice (ZH11) was grown in paddy fields until ripening, and various tissues were sampled for RNA analysis. The relative expression of *OsEnS-42* was measured by qRT-PCR using the reference gene eEF1a as a control. (**B**) Subcellular localization of OsEnS-42-GFP fusion protein in rice protoplasts. Confocal microscopy shows GFP signal (green) from OsEnS-42-GFP and mCherry nuclear marker signal (red). Yellow regions in merged images indicate nuclear co-localization. Bar = 10 μm.

**Figure 2 plants-14-02492-f002:**
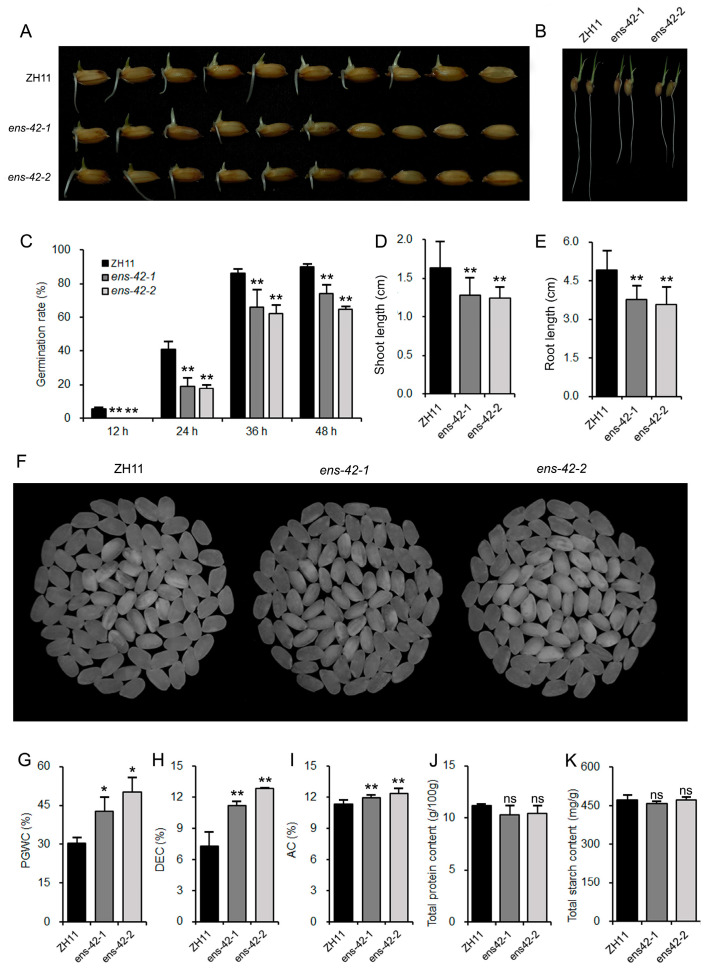
*OsEnS-42* regulates seed vigor and grain quality. (**A**) Comparison of seed vigor between ZH11 and KO lines after 48 h under germination conditions. (**B**) Comparison of shoot length and root length between ZH11 and KO lines after 96 h under germination conditions. (**C**) Comparison of germination rates between ZH11 and KO lines. Three biological replicates were conducted, with at least three parallel experiments per replicate, each containing 50 seeds. (**D**,**E**) Statistical analysis of shoot length and root length of ZH11 and KO lines. Root and shoot lengths were analyzed, with a minimum of 50 samples per genetic line. (**F**) Comparison images of chalkiness between ZH11 and KO lines. (**G**–**K**) Statistical analysis of percentage of grain with chalkiness (PGWC), degree of endosperm chalkiness (DEC), amylose content (AC), total protein content, and total starch content in ZH11 and KO lines. Data are means ± SD of three independent biological experiments. Statistical comparison was performed by one-sided *t*-test (ns: *p* > 0.05, * *p* < 0.05, ** *p* < 0.01).

**Figure 3 plants-14-02492-f003:**
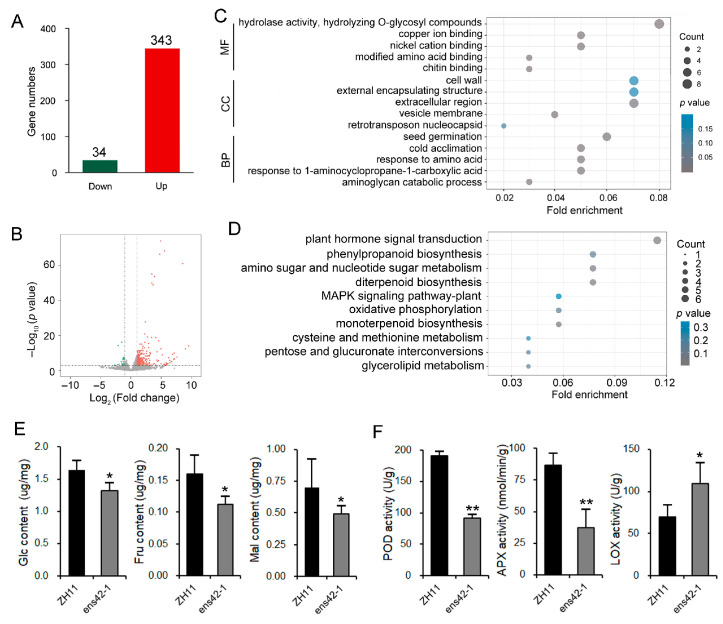
Transcriptome analysis of potential pathways through which *OsEnS-42* regulates seed vigor. (**A**) Number of DEGs in germinating seeds after 48 h under germination conditions. (**B**) Volcano plot of DEGs. (**C**) Gene Ontology (GO) enrichment analysis of DEGs. The top five pathways of the three main functional categories (BP, CC, and MF) were used for presentation. BP—biological process; CC—cellular component; MF—molecular function. (**D**) KEGG pathway enrichment analysis of DEGs. The top 10 enriched pathways were displayed. (**E**) Comparison of Glc, Fru, and Mal content between ZH11 and KO lines. Glc—glucose; Fru—fructose; Mal—maltose. (**F**) Activity comparison of POD, APX, and LOX in ZH11 and KO lines. POD—peroxidase; APX—ascorbate peroxidase; LOX—lipoxygenase. Data are means ± SD of three independent biological experiments. Statistical comparison was performed by one-sided *t*-test (* *p* < 0.05 and ** *p* < 0.01).

**Figure 4 plants-14-02492-f004:**
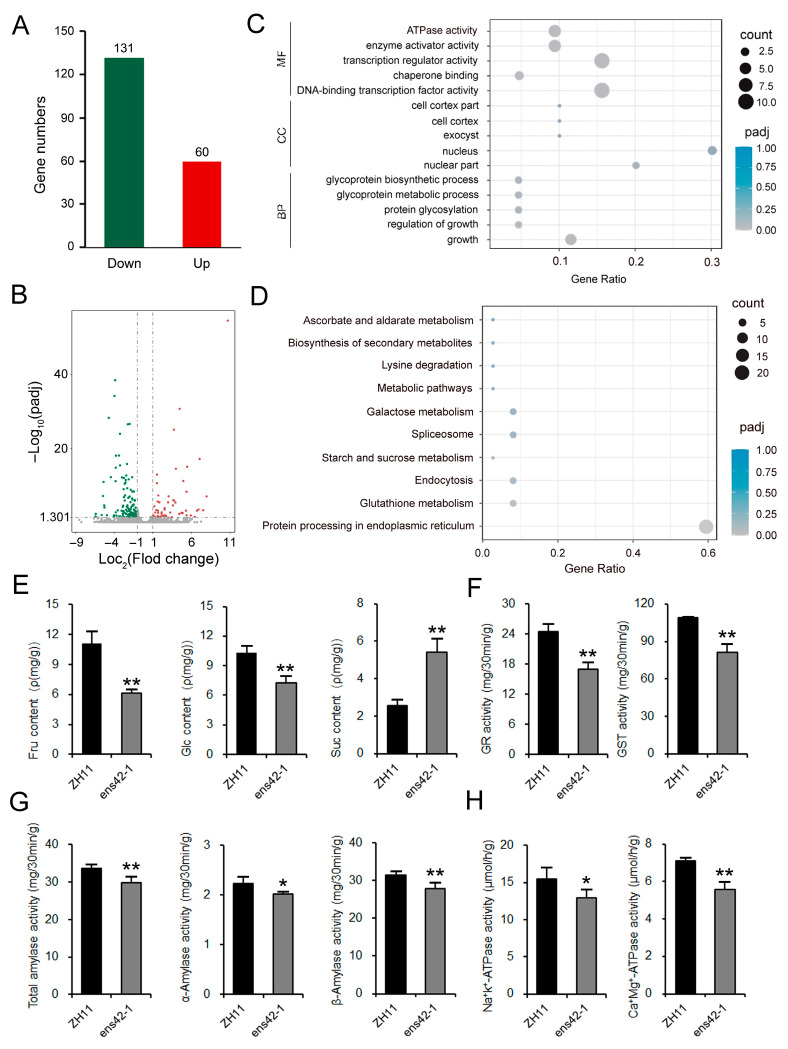
Transcriptome analysis of potential pathways through which *OsEnS-42* regulates grain quality. (**A**) Number of DEGs in panicles at one week after flowering. (**B**) Volcano plot of DEGs. (**C**) Gene Ontology enrichment analysis of DEGs. The top five pathways of the three main functional categories (BP, CC, and MF) are used for presentation. BP—biological process; CC—cellular component; MF—molecular function. (**D**) KEGG pathway enrichment analysis of DEGs. The top 10 enriched pathways are displayed. (**E**) Comparison of Fru, Glc, and Suc in ZH11 and KO lines. Fru—fructose; Glc—glucose; Suc—sucrose. (**F**) Activity comparison of GR and GST in ZH11 and KO lines. GR—glutathione reductase; GST—glutathione S-transferase. (**G**) Activity comparison of total amylase, α-amylase, and β-amylase in ZH11 and KO lines. (**H**) Activity comparison of Na^+^/K^+^-ATPase and Ca^2+^/Mg^2+^-ATPase in ZH11 and KO lines. Data are means ± SD of three independent biological experiments. Statistical comparison was performed by one-sided *t*-test (* *p* < 0.05 and ** *p* < 0.01).

**Figure 5 plants-14-02492-f005:**
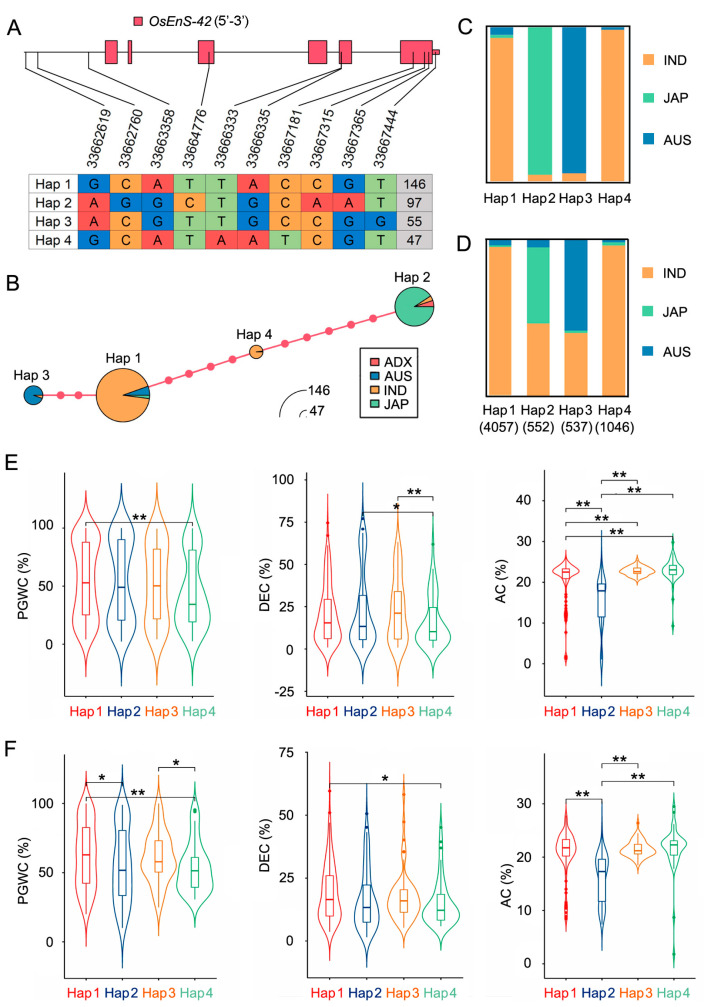
Alleles analysis of *OsEnS-42*. (**A**) Four haplotypes of *OsEnS-42* in the rice natural population. Ref—reference genome (*Nipponbare*). (**B**) Haplotype network of *OsEnS-42*. Circle size indicates sample number; dots on connecting lines represent mutation steps between haplotypes. (**C**) Distribution of four haplotypes in the core germplasm. (**D**) Distribution of four haplotypes in the 3K rice genome project. (**E**) Comparison of PGWC, DEC, and AC of accessions with four haplotypes in the Guangzhou ecological zone. PGWC—chalkiness; DEC—degree of endosperm chalkiness; AC—amylose content. (**F**) Comparison of PGWC, DEC, and AC of accessions with four haplotypes in the Yangjiang ecological zone. Statistical comparison was performed by one-sided *t*-test (* *p* < 0.05 and ** *p* < 0.01).

## Data Availability

The original contributions presented in this study are included in the article/Appendix A. The raw RNA-seq data are deposited in the NCBI database under accession number PRJNA1299026 (germinating seeds) and PRJNA1299049 (panicles).

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
