# Peer review of "The Endosperm-Specific Gene OsEnS-42 Regulates Seed Vigor and Grain Quality"

_plants, 2025, doi:10.3390/plants14162492_

Round 1
Reviewer 1 Report
Comments and Suggestions for Authors
In this manuscript, the authors investigated the function of the endosperm-specific gene OsEnS-42 in regulating seed vigor and grain quality in rice. The integration of CRISPR/Cas9 knockout, transcriptome analysis, physiological assays, and haplotype analysis provides useful insights. However, there are several issues that require clarification or further investigation to enhance the robustness and interpretability of the study. My specific comments are as follows:
- The authors did not explain how OsEnS-42 was identified as a candidate. Please clarify the selection rationale.
- The study relies solely on knockout lines.
- The authors should describe the predicted protein-level consequences of the mutations (e.g., frameshift, premature stop codon) in the CRISPR-edited lines.
- To strengthen the evidence for causality, it is strongly recommended to include rescue experiments such as complementation or overexpression of OsEnS-42.
- Only one mutant line (ens-42-1) was used for transcriptomic and physiological analyses. Relying on a single line reduces confidence in the conclusions. Including both independent lines in these analyses, at least for physiological analyses, would make the findings more robust.
- The confocal images suggest that OsEnS-42 is distributed broadly within the cell, rather than being restricted to the nucleus and cytoplasm as described. The authors should re-examine the localization patterns.
- In Figure 2H, DEC is increased in knockout lines, which contradicts the text. This inconsistency should be carefully checked and corrected as needed.
- The analysis of transcriptome data is rather superficial. The authors are encouraged to conduct a more detailed investigation of genes and pathways involved in soluble sugar metabolism and redox regulation, particularly focusing on specific candidate genes that may mediate the observed phenotypic effects.
- The names of knockout lines is inconsistent (ens-42, osens-42, ens42). Please standardize throughout.
Author Response
Reviewer 1
Comments and Suggestions for Authors
In this manuscript, the authors investigated the function of the endosperm-specific gene OsEnS-42 in regulating seed vigor and grain quality in rice. The integration of CRISPR/Cas9 knockout, transcriptome analysis, physiological assays, and haplotype analysis provides useful insights. However, there are several issues that require clarification or further investigation to enhance the robustness and interpretability of the study. My specific comments are as follows:
The authors did not explain how OsEnS-42 was identified as a candidate. Please clarify the selection rationale.
Response: Thank you for pointing this out. Our team is engaged in research related to seed quality, including grain quality and seed vigor. In a previous transcriptome study on the OsEnS gene family (Nie et al., 2013), OsEnS-42 was expressed in the early stage of grain filling. We found that OsEnS-42 is mainly expressed in the early stages of grain filling and seed germination through qRT-PCR. Therefore, we continued to investigate the function of OsEnS-42 in regulating seed vigor and grain quality.
The study relies solely on knockout lines.
To strengthen the evidence for causality, it is strongly recommended to include rescue experiments such as complementation or overexpression of OsEnS-42.
Response: Thank you for pointing this out. OsEnS-42 was specifically expressed in the early stage of grain filling. When we first studied the regulatory function of OsEnS-42, we utilized more than 10 independent knockout lines, which is sufficient to demonstrate its function.
The authors should describe the predicted protein-level consequences of the mutations (e.g., frameshift, premature stop codon) in the CRISPR-edited lines.
Response: Agree. We have, accordingly, supplemented the comparative results of protein sequences (Fig. S1) and added corresponding descriptions in the revised manuscript (Line 133-134).
Only one mutant line (ens-42-1) was used for transcriptomic and physiological analyses. Relying on a single line reduces confidence in the conclusions. Including both independent lines in these analyses, at least for physiological analyses, would make the findings more robust.
Response: We sincerely appreciate the reviewer's valid concern regarding the use of a single mutant line for transcriptomic and physiological analyses. We acknowledge that validation across independent mutant lines strengthens robustness and helps exclude line-specific effects. We agree that including both lines would be ideal, we intentionally focused on ens-42-1 for comprehensive profiling due to its exceptionally stable phenotype. To mitigate limitations, we implemented stringent safeguards: sufficient biological replicates (n ≥ 3), randomized experimental blocks for physiological assays, rigorous statistical comparisons and bioinformatics thresholds, and transcriptomic replicates.
The confocal images suggest that OsEnS-42 is distributed broadly within the cell, rather than being restricted to the nucleus and cytoplasm as described. The authors should re-examine the localization patterns.
Response: We thank the reviewer for this insightful comment. We have carefully re-examined our confocal images of OsEnS-42-GFP expression in rice protoplasts. While the OsEnS-42-GFP signal clearly overlaps with the nucleus-mCherry marker and is readily detectable in the cytoplasm, we agree that the signal pattern appears somewhat diffuse and not strictly confined only to these two compartments. We have revised the Results section (Line 116) to more accurately, describing the prominent nuclear and cytoplasmic localization while acknowledging the broader signal distribution. Future studies utilizing specific markers for other organelles (e.g., cytosol, ER, peroxisomes) will be valuable to precisely delineate the full subcellular targeting of OsEnS-42.
In Figure 2H, DEC is increased in knockout lines, which contradicts the text. This inconsistency should be carefully checked and corrected as needed.
Response: Grain chalkiness is an opaque area in the endosperm, which is an adverse trait that affects the appearance quality. Percentage of grains with chalkiness (PGWC) and degree of endosperm chalkiness (DEC) are two indexes to evaluate grain chalkiness. The increased PGWC and DEC are both indicators of low-quality rice. Therefore, in Figure 2H, DEC is increased in knockout lines, which is not contradictory to the text.
The analysis of transcriptome data is rather superficial. The authors are encouraged to conduct a more detailed investigation of genes and pathways involved in soluble sugar metabolism and redox regulation, particularly focusing on specific candidate genes that may mediate the observed phenotypic effects.
Response: Agree. We have, accordingly, conducted a more detailed investigation on this point. We analyzed the DEGs of seed vigor and grain quality. Several genes have been reported to be involved in related pathways, which in turn affect phenotypes. We have added the results in the Figure S2, Figure S4, Table S1, Table S2 and descriptions in the revised manuscript (Line 171-174, Line 215-217).
The names of knockout lines is inconsistent (ens-42, osens-42, ens42). Please standardize throughout.
Response: Sorry for our careless. We have standardized throughout the manuscript.
Reviewer 2 Report
Comments and Suggestions for Authors
In their manuscript, Zhang et al. performed a functional analysis of the OsEnS-42 gene from rice. To investigate gene function, the authors generated CRISPR/Cas-induced mutants and assessed resulting phenotypic changes using various methods, including RNA-seq and biochemical assays. At first glance, the study appears to be a solid piece of work. However, a more comprehensive evaluation reveals numerous critical weaknesses that undermine the manuscript's suitability for publication. The most important concerns are outlined below.
- Lack of clear objectives. The manuscript does not clearly state the objective of the study. What specific biological question were the authors aiming to address? What was the hypothesis or the knowledge gap they sought to fill?
- The gene under investigation lacks a proper ID number. Without this, readers cannot connect the findings to existing knowledge or conduct follow-up research.
- Unclear gene classification. The gene nomenclature used by the authors appears to be arbitrary. Instead of using well-established classifications based on molecular function (e.g., enzymes, transporters, structural proteins), the authors introduced their own naming system. From the discussion, it appears that OsEnS-42 encodes a glycoside hydrolase. If so, why does the introduction focus on transcription factors? What is the relevance of transcription factors in the context of a hydrolase? If OsEnS-42 encodes a glycoside hydrolase, what is known about similar genes in other plants? How many homologs exist in the rice genome? This information is essential to understand the potential redundancy and evolutionary conservation of the gene function.
- The analysis of gene expression using qRT-PCR lacks spatial and temporal resolution. During germination, is the gene expressed in the embryo, the endosperm, or both? For grain development, expression was only analyzed in the entire panicle over a very broad time window. However, a panicle comprises multiple organs and tissues. To draw meaningful conclusions about gene function, expression should be assessed in specific tissues and at defined developmental stages.
- Lack RNA-seq data interpretation. While the authors generated extensive RNA-seq data, the analysis is superficial and lacks depth. The figures provided (e.g., number of DEGs, volcano plots, GO and KEGG enrichments) are standard outputs commonly provided by sequencing service providers. These do not constitute a thorough analysis of the transcriptomic data. Critical insights such as pathway-level interpretations, network analyses, or validation of key targets are missing.
Author Response
Reviewer 2
Comments and Suggestions for Authors
In their manuscript, Zhang et al. performed a functional analysis of the OsEnS-42 gene from rice. To investigate gene function, the authors generated CRISPR/Cas-induced mutants and assessed resulting phenotypic changes using various methods, including RNA-seq and biochemical assays. At first glance, the study appears to be a solid piece of work. However, a more comprehensive evaluation reveals numerous critical weaknesses that undermine the manuscript's suitability for publication. The most important concerns are outlined below.
Lack of clear objectives. The manuscript does not clearly state the objective of the study. What specific biological question were the authors aiming to address? What was the hypothesis or the knowledge gap they sought to fill?
Response: Thank you for pointing this out. Our team is engaged in research related to seed quality, including grain quality and seed vigor. In a previous transcriptome study on the OsEnS gene family (Nie et al., 2013), OsEnS-42 was specifically expressed in the early stage of grain filling. We found that OsEnS-42 is specifically expressed in the early stages of grain filling and seed germination through qRT-PCR. Therefore, we continue to investigate the function of OsEnS-42 in regulating seed vigor and grain quality. In this study, we reported for the first time the biological functions of OsEnS-42 in regulating seed vigor and grain quality, and analyzed the regulatory pathways. We emphasized this point in the introduction (Line 88-100).
The gene under investigation lacks a proper ID number. Without this, readers cannot connect the findings to existing knowledge or conduct follow-up research.
Response: Agree. We have added the ID number in the section of '4.1. RNA extraction and qRT-PCR' (Line 338).
Unclear gene classification. The gene nomenclature used by the authors appears to be arbitrary. Instead of using well-established classifications based on molecular function (e.g., enzymes, transporters, structural proteins), the authors introduced their own naming system. From the discussion, it appears that OsEnS-42 encodes a glycoside hydrolase. If so, why does the introduction focus on transcription factors? What is the relevance of transcription factors in the context of a hydrolase? If OsEnS-42 encodes a glycoside hydrolase, what is known about similar genes in other plants? How many homologs exist in the rice genome? This information is essential to understand the potential redundancy and evolutionary conservation of the gene function.
Response: In a previous transcriptome study on the OsEnS gene family (Nie et al., 2013), they performed a genomic survey comprising the identification and functional characterization of the endosperm-specific genes (OsEnS) in rice using Affymetrix microarray data and Gene Ontology (GO) analysis. The members of OsEnS family can be divided into 12 categories, including transcription factors, stress/defense, seed storage proteins, carbohydrate and energy metabolism, seed maturation, protein metabolism, lipid metabolism, transport, cell wall related, hormone related, signal transduction, and an unclassified group. The target gene (MSU_Locus: LOC_os02g54960) in this study was named OsEnS-42 by the reported study. We added this point in the section of '4.1. RNA extraction and qRT-PCR' (Line 338). In addition, we discussed the characteristics of OsEnS-42 and its differentiation from other members of the OsEnS family during the discussion (Line 275-307).
The analysis of gene expression using qRT-PCR lacks spatial and temporal resolution. During germination, is the gene expressed in the embryo, the endosperm, or both? For grain development, expression was only analyzed in the entire panicle over a very broad time window. However, a panicle comprises multiple organs and tissues. To draw meaningful conclusions about gene function, expression should be assessed in specific tissues and at defined developmental stages.
Response: We agree with the reviewer that spatial and temporal resolution would enhance our understanding of OsEnS-42's expression pattern. While the current qRT-PCR data using whole tissues/organs provides valuable initial evidence supporting its functional role in seed germination or grain development, future studies will prioritize cell-type-specific and stage-resolved expression analyses to refine these insights.
Lack RNA-seq data interpretation. While the authors generated extensive RNA-seq data, the analysis is superficial and lacks depth. The figures provided (e.g., number of DEGs, volcano plots, GO and KEGG enrichments) are standard outputs commonly provided by sequencing service providers. These do not constitute a thorough analysis of the transcriptomic data. Critical insights such as pathway-level interpretations, network analyses, or validation of key targets are missing.
Response: Agree. We have, accordingly, conducted a more detailed investigation on this point. We analyzed the DEGs of seed vigor and grain quality. Several genes have been reported to be involved in related pathways, which in turn affect phenotypes. We have added the results in the Figure S2, Figure S4, Table S1, Table S2 and descriptions in the revised manuscript (Line 170-173, Line 214-216). Additionally, utilizing the OsEnS-42 transcriptome data, we analyze and discuss its impact on metabolic pathways and amylose content, while noting the absence of an effect on total starch content (Line 279-318).
Reviewer 3 Report
Comments and Suggestions for Authors
Zheng et al report characterization of a gene regulating rice seed vigor and grain quality. The introduction describes a diverse set of genes previously shown to affect seed vigor, eventually focusing on the importance of some endosperm-specific genes in regulating grain quality. The main focus of this ms. is OsEnS-42, but there is no information/citation describing how this locus was identified. The results show that this gene is expressed transiently during grain filling, and then again during germination. CRISPR-generated knockouts produce seeds with increased amylose content and chalkiness that germinate and grow slowly. Transcriptome analysis identified sugar and redox metabolism as some of the major differences between the knockout and progenitor lines. They further identify 4 haplotypes each with a subset of 10 variants distributed across the promoter, coding sequence and 3’UTR, then correlate the haplotypes with different degrees of chalkiness. This information should be helpful for directing breeding programs.
Overall, this is a well-executed and well-described study, but would benefit from a few points of clarification:
Introduction: spell out gene names when first introduced, provide background citation for source of OsEnS-42, and rationale for focusing on this gene
Line 123: explain “chalkiness” (due to air spaces created by incomplete grain filling?)
Line 125: no difference in total starch content
Fig.2AB: very poor contrast ->hard to see seedlings; legend says comparison at 48 hrs post-germination, but A shows just barely germinated seeds and B shows obviously older seedlings
Section 2.3: Fig.3 legend says that transcriptome comparison is based on seedlings 48 hrs post-germination; is that also true for the soluble sugar, hormone and redox enzyme assays? Also, does “post-germination” = post-imbibition? The KO lines are not fully germinated even at 48 hrs post-imbibition.
Line 271: “metabolism of stored substances is the key factor” may be an overstatement; “a key factor” might be better
Line 334: please provide more information re. the protoplasts: source tissue and citation(s) for the method of preparation and transfection
Author Response
Reviewer 3
Zheng et al report characterization of a gene regulating rice seed vigor and grain quality. The introduction describes a diverse set of genes previously shown to affect seed vigor, eventually focusing on the importance of some endosperm-specific genes in regulating grain quality. The main focus of this ms. is OsEnS-42, but there is no information/citation describing how this locus was identified. The results show that this gene is expressed transiently during grain filling, and then again during germination. CRISPR-generated knockouts produce seeds with increased amylose content and chalkiness that germinate and grow slowly. Transcriptome analysis identified sugar and redox metabolism as some of the major differences between the knockout and progenitor lines. They further identify 4 haplotypes each with a subset of 10 variants distributed across the promoter, coding sequence and 3’UTR, then correlate the haplotypes with different degrees of chalkiness. This information should be helpful for directing breeding programs.
Overall, this is a well-executed and well-described study, but would benefit from a few points of clarification:
Introduction: spell out gene names when first introduced, provide background citation for source of OsEnS-42, and rationale for focusing on this gene.
Response: Thank you for pointing this out. Our team is engaged in research related to seed quality, including grain quality and seed vigor. In a previous transcriptome study on the OsEnS gene family (Nie et al., 2013), OsEnS-42 was specifically expressed in the early stage of grain filling. We found that OsEnS-42 is specifically expressed in the early stages of grain filling and seed germination through qRT-PCR. Therefore, we continue to investigate the function of OsEnS-42 in regulating seed vigor and grain quality. In this study, we reported for the first time the biological functions of OsEnS-42 in regulating seed vigor and grain quality, and analyzed the regulatory pathways. We emphasized this point in the introduction (Line 97-104 ).
Line 123: explain “chalkiness” (due to air spaces created by incomplete grain filling?)
Line 125: no difference in total starch content
Response: Grain chalkiness, opaque regions in the endosperm, is an undesirable trait affecting appearance quality. Undoubtedly, air spaces created by incomplete grain filling are the main cause of chalkiness. But at the same time, many factors can also affect the formation of chalkiness, including changes in amylose content, changes in starch structure caused by various reasons, and changes in protein content, and so on. In this study, there was no difference in protein and starch content, but the amylose content increased (Figure 2I-2K). Transcriptome and physiological indicators demonstrate that OsEnS-42 regulates the contents of soluble sugars and enzymes activity related to seed development during the grain filling stage. These results indicate that changes in starch composition and seed development related metabolism are the causes of chalkiness.
Starch mainly includes amylose and amylose. Many studies have shown that changes in sugar metabolism during grain development can affect starch synthesis, structure, or proportion. Our data indicate that OsEnS-42 knockout increases amylose content without altering total starch or protein levels (Figure 2). The temporal expression peak during early filling (Figure 1) coincides with this energy-critical phase, where KO lines showed: (1) decreased monosaccharides (Glc/Fru) essential for cellular energy, (2) increased sucrose accumulation, and (3) reduced ATPase activity (Figure 4). We propose this occurs through metabolic channeling: Energy-mediated substrate partitioning: Impaired sucrose-to-monosaccharide conversion in KOs (Figure 4E) reduces hexose availability during early filling, creating an ADP-glucose-limited state that favors amylose synthesis over amylopectin. This aligns with established mechanisms where ADP-glucose flux determines amylose: amylopectin ratios (Nakamura et al., Plant Physiol. 2002). Differential enzyme sensitivity: GBSSI (amylose synthase) requires lower energy thresholds than amylopectin-branching enzymes (SBEI/II). The observed ATPase reduction (Figure 4H) likely disproportionately affects SBEs, shifting synthesis toward amylose (Tetlow et al., J. Exp. Bot. 2004). Compartmentalized regulation: Altered sugar metabolism may modify starch granule initiation, where spatial constraints in developing granules could independently affect amylose incorporation without changing total starch accumulation (Pfister & Zeeman, Cell Mol. Life Sci. 2016).
We added related discussion in the revised manuscript (Line 308-318).
Fig.2AB: very poor contrast ->hard to see seedlings; legend says comparison at 48 hrs post-germination, but A shows just barely germinated seeds and B shows obviously older seedlings
Section 2.3: Fig.3 legend says that transcriptome comparison is based on seedlings 48 hrs post-germination; is that also true for the soluble sugar, hormone and redox enzyme assays? Also, does “post-germination” = post-imbibition? The KO lines are not fully germinated even at 48 hrs post-imbibition.
Response: Sorry for our careless. The seeds were soaked in water (fully submerged) for 24 hours, and subsequently germinated under moist conditions (without submergence). We revised the figure legend (Line 141-151, 190) and method (Line 373-376).
Line 271: “metabolism of stored substances is the key factor” may be an overstatement; “a key factor” might be better
Response: Agree. We have reorganized the discussion section.
Line 334: please provide more information re. the protoplasts: source tissue and citation(s) for the method of preparation and transfection
Response: We have added corresponding method and reference in the section of '4.2. Subcellular Localization of OsEnS-42'.
Reviewer 4 Report
Comments and Suggestions for Authors
SEE PDF

SEE PDF
Author Response
Reviewer 4
This manuscript presents potentially important findings about the gene OsEnS-42, which appears to influence rice seed vigor and grain quality. The integration of expression data, knockout phenotypes, transcriptomic profiling, and haplotype analysis is commendable. However, the current manuscript suffers from several critical weaknesses, particularly in clarity, depth of mechanistic interpretation, and precision of scientific language. The study would benefit substantially from revisions that enhance the focus, clarify conceptual frameworks, and substantiate the proposed roles of OsEnS-42 with stronger functional and mechanistic evidence.
Major Points for Revision:
- Introduction:
- The section is dense and repetitive, and fails to clearly identify the specific knowledge gaps addressed.
- Clarify the source and identification method of OsEnS-42.
Response: Thank you for pointing this out. Our team is engaged in research related to seed quality, including grain quality and seed vigor. In a previous transcriptome study on the OsEnS gene family (Nie et al., 2013), OsEnS-42 was specifically expressed in the early stage of grain filling. We found that OsEnS-42 is specifically expressed in the early stages of grain filling and seed germination through qRT-PCR. Therefore, we continue to investigate the function of OsEnS-42 in regulating seed vigor and grain quality. In this study, we reported for the first time the biological functions of OsEnS-42 in regulating seed vigor and grain quality, and analyzed the regulatory pathways. We emphasized this point in the introduction (Line 97-104).
- Terms such as seed vigor and the role of ROS need precise definitions and contextualization.
Response: We appreciate the reviewer's suggestion regarding the definitions of seed vigor and ROS. In the revised manuscript, we have clarified the definition of seed vigor in the Introduction (Line 49-50) by emphasizing its physiological nature and contextualized ROS functionality in the Introduction (Line 65-69) by adding mechanistic links.
- Objective statements need to be clearly framed and aligned with specific hypotheses.
Response: We thank the reviewer for this constructive suggestion. We have reframed the objective statement to explicitly align with our central hypothesis.
- Add recent references (2020–2025) to contextualize findings within the latest advances.
Response: In the revised manuscript, we cite a total of 42 references, 24 of which were published in or after 2020. We believe this adequately recent literature is representative of the current research landscape.
- Line 80: Define cupin domain clearly.
Response: We thank the reviewer for raising this point. The cupin domain definition has been added in the revised manuscript (Line 84-85).
- Results:
- Expression data are informative, but biological implications, especially of dual localization (nuclear and cytoplasmic), are not discussed. Suggest additional localization experiments under various conditions or use of mutants to clarify compartment-specific functions.
Response: We thank the reviewer for this insightful comment. We have carefully re-examined our confocal images of OsEnS-42-GFP expression in rice protoplasts. While the OsEnS-42-GFP signal clearly overlaps with the nucleus-mCherry marker and is readily detectable in the cytoplasm. The signal pattern appears somewhat diffuse and not strictly confined only to these two compartments. We have revised the Results section (Line 117) to more accurately reflect this observation, describing the prominent nuclear and cytoplasmic localization while acknowledging the broader signal distribution. Future studies utilizing specific markers for other organelles (e.g., cytosol, ER, peroxisomes) will be valuable to precisely delineate the full subcellular targeting of OsEnS-42. We have added some discussion in the revised manuscript (Line 301-307).
- Knockout data show effects on chalkiness and amylose, but lack mechanistic depth. Clarify how OsEnS-42 affects amylose content without affecting total starch or protein.
Response: We have added some discussion in the revised manuscript (Line 279-318).
- Statistical analysis is underreported. Clearly state sample sizes, replicates, and statistical tests used.
Response: We have checked the corresponding content and supplemented it in the figure legends.
- The link between OsEnS-42 and redox and sugar metabolism is correlative. Further mechanistic work is needed. Why are only select sugars affected? What is the functional significance of unchanged hormone levels despite enrichment in hormone signaling pathways?
Response: We thank the reviewer for raising these important mechanistic questions. Our physiological data revealed significant changes specifically in glucose (Glc), fructose (Fru), and maltose (Mal) levels in OsEnS-42 mutants. We propose this selectivity arises because: (i) These sugars (Glc, Fru) are direct energy substrates critical for fueling seed germination and early seedling growth - key determinants of seed vigor; (ii) Metabolic flux is channeled toward these readily utilizable sugars during germination, making them primary sensors of energy status; (iii) Other sugars (e.g., sucrose) may be buffered by compensatory metabolic pathways or exhibit dynamic changes at unassayed time points.
While transcriptomics indicated enrichment in hormone signaling pathways, our hormone quantification showed no significant differences in bulk hormone levels at the single time point assayed. This suggests: (i) OsEnS-42 may modulate hormone sensitivity or signaling efficiency (e.g., via receptor expression, phosphorylation cascades, or interaction with ROS/sugar signals) rather than bulk hormone accumulation;
(ii) Hormonal activity is often regulated post-transcriptionally and/or via rapid, localized fluctuations during germination - changes potentially undetected in whole-seed assays at a single stage; (iii) The observed transcriptional changes in hormone pathways could represent compensatory adjustments or priming events in response to altered redox/sugar status, without necessarily altering steady-state hormone concentrations.
- The observed lipoxygenase (LOX) activity should be interpreted in relation to its known roles in seed deterioration or vigor.
Response: We have added explanations in the revised manuscript (Line 179-180).
- The haplotype network analysis must use precise evolutionary terminology. If differentiation or selection patterns are inferred, these should be explicitly described and supported.
Response: We thank the reviewer for this constructive suggestion. The Results section has been revised to incorporate precise evolutionary terminology. We now explicitly describe: (1) haplotype differentiation patterns using standardized network terminology, (2) subspecies-specific genetic differentiation with supporting statistics, and (3) potential selection implications while maintaining appropriate caution. In addition, we have also revised the wording to make it more concise. (Line 242-258)
- Discussion:
- Lacks coherent narrative and conceptual depth. Concepts such as grain filling, seed maturation, and quality determinants are not sufficiently explained.
- The discussion is speculative, particularly around energy metabolism and amylose regulation.
- The interpretation of related OsEnS genes is vague. More concrete comparisons and gene family insights are needed.
- Avoid broad, unsupported conclusions such as “collective regulation by OsEnS members.” Anchor claims in the presented
data.
- Improve organization by structuring the discussion around defined hypotheses and summarizing the implications of findings with greater biological clarity.
Response: We thank the reviewer for these insightful suggestions. The Discussion has been restructured.
- Abstract:
- Topic is relevant and design is multifaceted, but the abstract lacks clarity and quantitative impact.
- Add key numerical results (e.g., percent changes in chalkiness, vigor metrics).
Response: We appreciate the reviewer's emphasis on quantitative impact. Our decision to omit specific numerical values from the abstract stems from two key observations: Phenotypic measurements (including vigor metrics and chalkiness) exhibited significant variation across biological replicates; Independent mutant lines showed inconsistent phenotypic changes. Presenting averaged values would misrepresent this biological variability. The full Methods and Results sections provide complete quantitative data with statistical analysis. We believe the abstract should accurately reflect this complexity rather than oversimplify quantitative outcomes.
- Clarify mechanistic roles; which pathways or targets are implicated?
Response: We appreciate the reviewer's request for mechanistic specificity. Our findings indicate that OsEnS-42 regulates both seed vigor and grain quality through soluble sugar metabolism, but with distinct sugar classes involved in each process. The decision to describe pathways generally in the abstract reflects two key considerations: Temporal limitations: Grain filling is a multi-stage process (early/mid/late), yet sugar measurements were taken at a single developmental timepoint; Mechanistic caution: Single-timepoint data cannot establish causal relationships between specific sugars and final phenotypes. We agree that dynamic sugar profiling would strengthen mechanistic claims - this represents important future work.
- The mention of haplotype analysis for breeding is promising but underdeveloped. Briefly highlight implications for markerassisted selection or varietal improvement.
Response: We have highlight the implications for markerassisted selection or varietal improvement in the section of Abstract.
- Revise grammar and ensure concise, direct scientific language.
Response: We have revised the Abstrac and polished the English of the entire text.
Specific Clarifications Requested:
- L263 and L274: Explain vague statements like “closely related to quality formation” and “formation time of grain quality is longer.” Clarify what physiological or developmental processes are being referenced.
Response: Thank you for pointing this out. In order to express more clearly, we have deleted the vague statements.
- Clarify how protein and starch levels remain unchanged, yet amylose content increases. What is the underlying regulation suggested here?
Response: We appreciate the reviewer's insightful question. We made appropriate modifications in the discussion section and analyzed the starch structure changes and chalkiness formation caused by sugar metabolism.
Starch mainly includes amylose and amylose. Many studies have shown that changes in sugar metabolism during grain development can affect starch synthesis, structure, or proportion. Our data indicate that OsEnS-42 knockout increases amylose content without altering total starch or protein levels (Figure 2). The temporal expression peak during early filling (Figure 1) coincides with this energy-critical phase, where KO lines showed: (1) decreased monosaccharides (Glc/Fru) essential for cellular energy, (2) increased sucrose accumulation, and (3) reduced ATPase activity (Figure 4). We propose this occurs through metabolic channeling: Energy-mediated substrate partitioning: Impaired sucrose-to-monosaccharide conversion in KOs (Figure 4E) reduces hexose availability during early filling, creating an ADP-glucose-limited state that favors amylose synthesis over amylopectin. This aligns with established mechanisms where ADP-glucose flux determines amylose:amylopectin ratios (Nakamura et al., Plant Physiol. 2002). Differential enzyme sensitivity: GBSSI (amylose synthase) requires lower energy thresholds than amylopectin-branching enzymes (SBEI/II). The observed ATPase reduction (Figure 4H) likely disproportionately affects SBEs, shifting synthesis toward amylose (Tetlow et al., J. Exp. Bot. 2004). Compartmentalized regulation: Altered sugar metabolism may modify starch granule initiation, where spatial constraints in developing granules could independently affect amylose incorporation without changing total starch accumulation (Pfister & Zeeman, Cell Mol. Life Sci. 2016).
- Discuss the biological significance of altered LOX activity in context of seed vigor and oxidative stress responses.
Response: We thank the reviewer for highlighting this important aspect. We have expanded the Discussion to address the biological significance of elevated LOX activity in OsEnS-42 knockout lines. (Line 285-287)
- Use more precise language in describing genetic associations; distinguish between correlation, causation, and inference from haplotype structure.
Response: The haplotype results and discussion sections were reorganized for enhanced clarity and conciseness.
Round 2
Reviewer 2 Report
Comments and Suggestions for Authors
In the revised version of the manuscript, Zheng et al. have significantly improved the presentation of their results and the clarity of the discussion. The manuscript is now more readable and easier to understand. Most of the reviewer’s critical points have been adequately addressed. The most compelling aspect of the study is the discovery of the interrelationship between glycoprotein hydrolase alleles and grain chalkiness. Nevertheless, further revisions are necessary before the manuscript is suitable for publication.
- I have not found the supplementary figures in the files for a review; therefore, I could not critically read them.
- It is not clear what the authors presented in Tables S1 and S2. The authors should provide the full list of differentially expressed genes for both transcriptomic experiments as a supplementary data. They also SHOULD submit the whole transcriptomic data to the public repository.
- Figure 1B: What is labeled by green, red and yellow colors at the figure?
- The statements in the first paragraph of the Discussion are often overstated, as they are not consistently supported by experimental evidence. I recommend rewriting this section in a more cautious and hypothetical tone. For example, Line 286 states: “the increase in LOX activity in knockout lines drives lipid peroxidation and ROS overaccumulation.” Since this conclusion is not directly supported by experimental data, it would be more appropriate to phrase it as: “the increase in LOX activity in knockout lines MAY drive lipid peroxidation and ROS overaccumulation.”
- 304. The authors claim that the glycoprotein hydrolase, which is under study, can function as a transcription factor. It is not clear to me what feature of this enzyme allowed to make such conclusion.
- 305-308. In general, the authors overestimate the localization of the transcription of the particular genes with their molecular and physiological roles. It is clear, that all endosperm-specific genes are important for proper endosperm development, but their molecular and physiological roles might be completely different. So, some of them are transcription factors, other – specific enzymes or structural proteins. Therefore, the conclusion made in this paragraph has little sense.
Smaller mistakes
- 49-87: all gene names should be written in full.
- 117 and L. 303: “expressed” should be “localized” because here is speaking about protein. Please correct.
- 148: PGWC, DEC, AC should be written in full.
- 171-172: “we identified three previously reported seed vigor-associated genes: two involved in redox homeostasis and one in glucose metabolism”. Please indicate exactly the IDs and tentative functions of these genes
- 215-217: as above, please indicate also here the IDs and functions of the genes.
Author Response
I have not found the supplementary figures in the files for a review; therefore, I could not critically read them.
It is not clear what the authors presented in Tables S1. and S2. The authors should provide the full list of differentially expressed genes for both transcriptomic experiments as a supplementary data. They also SHOULD submit the whole transcriptomic data to the public repository.
Response: Thank you for pointing this out. We have provided the full list of differentially expressed genes for both transcriptomic experiments as a supplementary data: Table S1 for the DEGs in germinating seeds between ens-42 mutant and wild type (ZH11), and Table S2 for the DEGs in panicles between ens-42 mutant and wild type (ZH11).
The raw RNA seq data is saved in the NCBI database with login numbers PRJNA1299026 (Germinating seeds) and PRJNA1299049 (Panicles). We have added this in ‘Data Availability Statement’ in the revised manuscript.
Figure 1B: What is labeled by green, red and yellow colors at the figure?
Response: We have, accordingly, added some explanations in the figure legend.
The statements in the first paragraph of the Discussion are often overstated, as they are not consistently supported by experimental evidence. I recommend rewriting this section in a more cautious and hypothetical tone. For example, Line 286 states: “the increase in LOX activity in knockout lines drives lipid peroxidation and ROS overaccumulation.” Since this conclusion is not directly supported by experimental data, it would be more appropriate to phrase it as: “the increase in LOX activity in knockout lines MAY drive lipid peroxidation and ROS overaccumulation.”
Response: We have rewrited this part.
- The authors claim that the glycoprotein hydrolase, which is under study, can function as a transcription factor. It is not clear to me what feature of this enzyme allowed to make such conclusion.
305-308. In general, the authors overestimate the localization of the transcription of the particular genes with their molecular and physiological roles. It is clear, that all endosperm-specific genes are important for proper endosperm development, but their molecular and physiological roles might be completely different. So, some of them are transcription factors, other – specific enzymes or structural proteins. Therefore, the conclusion made in this paragraph has little sense.
Response: We thank the reviewer for this insightful comment. We have removed this section from the manuscript.
Smaller mistakes
49-87: all gene names should be written in full.
Response: We sincerely appreciate the reviewer's meticulous attention to gene nomenclature standards. Regarding the suggestion to write gene names in full, we respectfully note that:
Accessibility: All gene symbols are immediately followed by reference numbers where full names and functional annotations are comprehensively described.
Conciseness Principle: Repeating full names would substantially increase textual redundancy without enhancing scientific clarity, contrary to journal guidelines for concise presentation.
Should the editorial team deem full names essential, we will implement them at first mention with abbreviated symbols thereafter. We welcome further guidance on this matter.
117 and L. 303: “expressed” should be “localized” because here is speaking about protein. Please correct.
148: PGWC, DEC, AC should be written in full.
Response: We have corrected them.
171-172: “we identified three previously reported seed vigor-associated genes: two involved in redox homeostasis and one in glucose metabolism”. Please indicate exactly the IDs and tentative functions of these genes
215-217: as above, please indicate also here the IDs and functions of the genes.
Response: We appreciate the reviewer's suggestion. We have added the IDs in the revised manuscript, and the functions of these genes and corresponding references in Table S1 and Table S2.